# IMPROVING SAMPLE EFFICIENCY IN OFF-POLICY RL WITH LOW-DIMENSIONAL POLICY REPRESENTATION

## ABSTRACT

Off-policy Reinforcement Learning (RL) is fundamental to realizing intelligent decision-making agents by trial and error. The most notorious issue of off-policy RL is known as *Deadly Triad*, i.e., Bootstrapping, Function Approximation, and Off-policy Learning. Despite recent advances in bootstrapping algorithms with better bias control, improvements in the latter two factors are relatively less studied. In this paper, we propose an efficient and general off-policy RL algorithm based on low-dimensional policy representations. Orthogonal to better bootstrapping, our improvement is two-fold. On the one hand, the policy representation serves as an additional input to the value function, allowing it to offer preferable function approximation with less interference and better generalization. On the other hand, the policy representation empowers off-policy RL methods to perform off-policy learning in a more sufficient manner. Specifically, we perform additional value learning for proximal historical policies along the learning process. This drives the value generalization from learned policies and in turn, leads to more efficient learning. We evaluate our algorithms on continuous control tasks and the empirical results demonstrate consistent improvements in terms of efficiency and stability.

## 1 INTRODUCTION

Off-policy Reinforcement Learning is an important branch of reinforcement learning that has attracted much attention thanks to its generality and application potential François-Lavet et al. (2018). In off-policy RL, three widely utilized techniques are: Bootstrapping, Function Approximation, and Off-policy Learning, collectively referred to as *Deadly Triad* Sutton (1988); Sutton & Barto (2018). Despite high-profile empirical successes, sample inefficiency and learning instability due to the Deadly Triad remain key issues of off-policy RL Achiam et al. (2019); Van Hasselt et al. (2018). This greatly limits the deployment of off-policy RL in real-world scenarios. In recent years, significant progress has been made by improving bootstrapping methods, resulting in more advanced off-policy RL algorithms Fujimoto et al. (2018); Kuznetsov et al. (2020); Lan et al. (2020); Chen et al. (2021); Liang et al. (2022). Nevertheless, the latter two factors (i.e., function approximation and off-policy learning) receive relatively less attention. Moreover, existing works usually only focus on one of these two factors, either how to design better function approximators from network structures (usually modeled by neural networks) Ota et al. (2020); Shah & Kumar (2021) or compensate for the discrepancy between the distributions of the policy of interest and the behavior policy Saglam et al. (2022); Kumar et al. (2020).

Different from previous works, our work attempts to improve both simultaneously to obtain a more advanced and general off-policy RL algorithm. Our idea is mainly inspired by the appealing characteristics of policy representation in value generalization Harb et al. (2020); Faccio et al. (2020); Tang et al. (2020); Raileanu et al. (2020); Sang et al. (2022). Specifically, policy parameters Faccio et al. (2020) and policy representations Tang et al. (2020) obtained by encoding the policy parameters are respectively used as inputs to the value function approximator to improve function approximation for on-policy RL algorithms. In the offline setting, the PVN (Policy Evaluation Networks) Harb et al. (2020) is proposed to approximate the expected return of multiple policies with policy representations obtained by policy fingerprints as input. Furthermore, towards obtaining preferable function approximation with better generalization across policies and tasks, Raileanu et al. Raileanu et al. (2020) and Sang et al. Sang et al. (2022) propose to use both a policy representation and a task representation as additional inputs to the value function approximator.

Essentially, the aforementioned works propose different policy representations to improve function approximation. Rather than limiting ourselves to function approximation, in this paper, we are dedicated to exploring the potential of policy representation for both function approximation and off-policy learning with respect to off-policy RL characteristics. This has not been discussed before. First of all, we propose a new Bellman operator with policy representations to characterize off-policy RL for the purpose of better value function approximation and generalization. Building upon the proposed Bellman operator, we further develop a **G**eneralized **O**ff-**P**olicy **E**valuation manner (**GOPE**) to improve sample efficiency. Moreover, we propose a simple and effective policy representation learning method, named **L**ayer-wise **P**ermutation-invariant **E**ncoder with **D**ynamic **M**asking (**LPE-DM**), which follows the characteristics of policy data itself for learning policy representations. To evaluate the effectiveness and generality of the proposed method, we present two practical implementations of our method based on TD3 Fujimoto et al. (2018) and SAC Haarnoja et al. (2018b). We evaluate them on six OpenAI continuous control tasks, and the empirical results indicate that our algorithm significantly outperforms the benchmarks in each tested environment.

We summarize our main contributions below: 1) We propose an efficient and general off-policy RL algorithm based on low-dimensional policy representations, which leverages value generalization among policies to improve the learning process. 2) We propose a new policy representation learning method for the effective encoding of policy networks. 3) Empirical results on popular OpenAI Gym control tasks demonstrate the consistent superiority of our algorithms in terms of efficiency and stability.

## 2 PRELIMINARY

### 2.1 REINFORCEMENT LEARNING

A Markov Decision Process (MDP) Puterman (2014) is usually defined by a five-tuple $\langle S, A, P, r, \gamma \rangle$, with the state space $S$, the action space $A$, the transition probability $P : S \times A \rightarrow \Delta(S)$, the reward function $r : S \times A \rightarrow \mathbb{R}$ and the discount factor $\gamma \in [0, 1)$. $\Delta(X)$ denotes the probability distribution over $X$. A stationary policy $\pi \in \Pi : S \rightarrow \Delta(A)$ is a mapping from states to action distributions, which defines how to behave under specific states. An agent interacts with the MDP at discrete timesteps by its policy $\pi$, generating trajectories with $s_0 \sim \rho_0(\cdot)$, $a_t \sim \pi(\cdot|s_t)$, $s_{t+1} \sim P(\cdot \mid s_t, a_t)$ and $r_{t+1} = r(s_t, a_t)$, where $\rho_0$ is the initial state distribution. The goal of an RL agent is to maximize the value defined as the expected cumulative discounted returns $\mathbb{E}_{s_0 \sim \rho_0}[\sum_{t=0}^{\infty} \gamma^t r_{t+1} \mid s_0]$. Given a policy $\pi$, the discounted state visitation distribution from initial states regarding $\rho_0$ is defined as $d^\pi(s)$. There exists an action-value function $Q^\pi(s, a) = \mathbb{E}_{s \sim d^\pi(s), a \sim \pi}[\sum_{t=0}^{\infty} \gamma^t r_{t+1} \mid s_0 = s, a_0 = a]$. We compute the action-value function through the Bellman operator, $\mathcal{T}^\pi Q^\pi(s, a) = \mathbb{E}_{s' \in S}[r + \gamma Q^\pi(s', \pi(s'))]$ Sutton & Barto (2018). The optimal action-value function $Q^*(s, a) = \max_{a \in A} Q^\pi(s, a)$ is obtained by the greedy actions of the corresponding policy.

### 2.2 OFF-POLICY RL

In deep RL, the action-value function is usually modeled by a differentiable function approximator $Q_\theta(s, a)$ with parameters $\theta$, commonly known as Q-network. $Q_\theta(s, a)$ is obtained by temporal difference (TD) learning Sutton (1988) based on the Bellman Equation:

$$Q_\theta(s, a) \leftarrow (r + \gamma Q_{\theta^-}(s', a')), \forall s, s' \in \mathcal{S}, a, a' \in \mathcal{A}, \tag{1}$$

where $Q_{\theta^-}$ is the target network for providing a fixed objective to the Q-network and ensuring stability in the updates. Typically, $Q_{\theta^-}$ is updated by some proportion $\tau$ at each time step $\theta^- \leftarrow \tau\theta + (1-\tau)\theta^-$, named soft-update. In continuous action spaces, it is intractable to obtain the maximum value of the action-value function Saglam et al. (2022). Thus, a separate network named the actor network $\pi_\phi$ is employed which selects actions on the observed states. $\pi_\phi$ with parameters $\phi$ is optimized by one-step gradient ascent over the policy gradient $\nabla_\phi J(\phi)$ following the policy gradient theorem Sutton & Barto (2018); Silver et al. (2014). In off-policy deep RL, the policy $\pi_\phi$ can be optimized using collected data that are not necessarily obtained under the current policy $\pi_\phi$, but from a *behavioral* policy $\pi_\beta$. In this case, the deterministic policy gradient and stochastic policy gradient are respectively:

$$\nabla_\phi J_{det}(\phi) = \mathbb{E}_{s \sim d^{\pi_\beta}(s)}\left[\nabla_a Q_\theta(s, a)|_{a=\pi_\phi(s)} \nabla_\phi \pi_\phi(s)\right], \tag{2}$$

$$\nabla_\phi J_{sto}(\phi) = \mathbb{E}_{s \sim d^{\pi_\beta}(s), a \sim \pi_\phi} \left[ Q_\theta(s, a) \nabla_\phi \log \pi_\phi(a \mid s) \right]. \tag{3}$$

The sample efficiency of off-policy RL is improved as they make use of any past experience Degris et al. (2012). This may be especially useful in some scenarios where it may be costly or dangerous to collect data using the learned policy.

## 3 OFF-POLICY RL WITH POLICY REPRESENTATION

Towards function approximation and off-policy learning, in this section, we first introduce the value generalization based on the newly proposed Bellman operator $\mathbb{T}^\pi$ (Sec. 3.1). Then, we propose a more general and sufficient manner to perform off-policy learning via policy representations (Sec. 3.2). Finally, we present in detail a practical implementation of the proposed algorithm (Sec. 3.3).

### 3.1 VALUE GENERALIZATION WITH POLICY REPRESENTATION

In general, off-policy RL algorithms learn the action-value function $Q^\pi$ through the Bellman operator $\mathcal{T}^\pi$: $\mathcal{T}^\pi Q^\pi(s, a) = \mathbb{E}_{s' \in S} \left[ r + \gamma Q^\pi(s', \pi(s')) \right]$. Thus, we present our algorithm starting from the Bellman operator. Following the definition of the Bellman operator, we propose a new Bellman operator $\mathbb{T}^\pi$ regarding policy representation as follows:

**Definition 3.1.** Let $\mathbb{T}^\pi : \mathbb{Q} \to \mathbb{Q}$ be the operator on the policy representation-based action-value function $\mathbb{Q}$. For a given policy $\pi \in \Pi$, $f(\pi)$ is a policy representation function that maps the policy $\pi$ to a low-dimensional policy representation $\chi_\pi \in \mathcal{X}$. $\forall s, s' \in S, a, a' \in A$, the new Bellman operator $\mathbb{T}^\pi$ is defined as:

$$\mathbb{T}^\pi \mathbb{Q}^\pi(s, a, \chi_\pi) = \mathbb{E}_{s' \in S} \left[ r + \gamma \mathbb{Q}^\pi(s', \pi(s'), \chi_\pi) \right]. \tag{4}$$

$\mathbb{T}^\pi$ is a recursive operator which satisfies the *compression map theorem* Sutton & Barto (2018) with a unique fixed-point $\mathbb{Q}^\pi$, denoted as $\mathbb{T}^\pi \mathbb{Q}^\pi = \mathbb{Q}^\pi$. Hence, for arbitrary policy $\pi \in \Pi$, we perform multiple Bellman operations on its initial action-value function $\mathbb{Q}_0^\pi$ to obtain the unique fixed-point $\mathbb{Q}^\pi$, i.e., convergence to the action-value function of the policy $\pi$.

The learning process of action-value function based on the two Bellman operators $\mathcal{T}^\pi$ and $\mathbb{T}^\pi$ can be expressed as $\lim_{n \to \infty} \mathcal{T}_n^\pi Q_0^\pi = Q^\pi$ and $\lim_{n \to \infty} \mathbb{T}_n^\pi \mathbb{Q}_0^\pi = \mathbb{Q}^\pi$, respectively. Somewhat naturally, the closer $Q_0^\pi$ and $\mathbb{Q}_0^\pi$ is to the true value $Q^\pi$, the smaller $n$ is, resulting in the higher efficiency of value learning. Thus, a key question is whether $\mathbb{Q}_0^\pi$ is closer to the true value $Q^\pi$ than $Q_0^\pi$. We first study the value approximation and value generalization in a two-policy case (i.e., $\pi_t, \pi_{t+1}$) where only the value of policy $\pi_t$ is approximated by $\mathbb{Q}_\theta$ with parameters $\theta$ as below:

**Definition 3.2** (Value Approximation with $\mathbb{T}^\pi$). We define a value learning process with $\mathbb{T}^\pi$, $\mathcal{P}_\pi$ : $\Theta \xrightarrow{\mathbb{T}^\pi} \hat{\Theta}$. Given a policy $\pi \in \Pi$, the action-value function $Q^\pi$ of $\pi$ can be approximated by $\mathbb{Q}_{\hat{\theta}}^\pi(\chi_\pi)$ and the value approximation distance is defined as $f_{\hat{\theta}}(\pi) = \|\mathbb{Q}_{\hat{\theta}}(\chi_\pi) - Q^\pi\|$.

**Theorem 3.3** (Value Generalization with $\mathbb{T}^\pi$). *Given consecutive policies $\pi_t$, $\pi_{t+1}$ in policy iteration process, if $f_{\hat{\theta}_t}(\pi_t) + f_{\hat{\theta}_t}(\pi_{t+1}) \leq \|Q^{\pi_t} - Q^{\pi_{t+1}}\|$, then $\|\mathbb{Q}_{\hat{\theta}_t}(\chi_{\pi_{t+1}}) - Q^{\pi_{t+1}}\| \leq \|\mathbb{Q}_{\hat{\theta}_t}(\chi_{\pi_t}) - Q^{\pi_{t+1}}\|$.*

Theorem 3.3 can be proved by the *Triangle Inequality*. Based on the Theorem 3.3, the value learning at policy $\pi_{t+1}$ starts from the generalized value estimation $\mathbb{Q}_{\hat{\theta}_t}(\chi_{\pi_{t+1}})$. Conversely, with respect to the action-value function without policy representation, the value learning at policy $\pi_{t+1}$ starts from the $Q_{\hat{\theta}_t}$ which is equivalent to $\mathbb{Q}_{\hat{\theta}_t}(\chi_{\pi_t})$. Thus, $\mathbb{T}^\pi$ with policy representation offers value generalization among the policy space. In particular, under bootstrapping-based TD learning, value generalization can be obtained from $\mathbb{Q}_{\hat{\theta}_t}(\chi_{\pi_{t+1}})$ and $\mathbb{Q}_{\hat{\theta}_t^-}(\chi_{\pi_{t+1}})$. Taking a further step from the two-policy case above, we recall the Generalized Policy Iteration (GPI) Sutton (1988) followed by most RL algorithms, the consecutive value approximation for the policies along the policy improvement path can be described as $\theta_{-1} \xrightarrow{\mathcal{P}_{\pi_0}} \theta_0 \xrightarrow{\mathcal{P}_{\pi_1}} \theta_1 \cdots$. The value learning from each $\pi_t$ and $\pi_{t+1}$ during GPI can be similarly considered as the two-policy case. Thus, importing policy representation into GPI-based off-policy RL can greatly improve value learning efficiency.

## 3.2 Generalized Off-policy Evaluation with Policy Representation

In this section, we mainly investigate *how to make use of policy representation to improve off-policy evaluation.* To the best of our knowledge, there is no related work to improve the off-policy evaluation process from the policy representation perspective. As introduced in Sec.2.2, the key characteristic of off-policy evaluation is that the value approximation is performed using the experiences collected by the encountered policies along the policy improvement path, which improves sample efficiency compared to on-policy evaluation. In order to further enhance the advantages of off-policy evaluation in terms of sample efficiency, we propose a generalized off-policy evaluation manner with policy representation. Specifically, unlike traditional off-policy evaluation which is limited to the current policy to be learned, we borrow policy representations to achieve off-policy evaluation of any historical policy using any historical transition experience. We refer to the generalized off-policy evaluation manner as GOPE.

**GOPE.** During the GPI ($\theta_{-1} \xrightarrow{\mathbb{T}_n^{\pi_0}} \theta_0, \cdots, \theta_t \xrightarrow{\mathbb{T}_n^{\pi_{t+1}}} \theta_{t+1}, \cdots$), for a value approximation process $\theta_t \xrightarrow{\mathbb{T}_n^{\pi_{t+1}}} \theta_{t+1}$, we perform off-policy learning of the historical policies, i.e., $\theta_t \xrightarrow{\mathcal{P}_{\pi \in \Pi_t^{GPI}}} \theta_t' \xrightarrow{\mathbb{T}_n^{\pi_{t+1}}} \theta_{t+1}$. $\Pi_t^{GPI}$ denotes the policy subset obtained along the GPI at the $t$-th iteration. Thanks to the off-policy learning manner provided by one-step TD estimation, $\forall \pi_i \in \Pi_t^{GPI}$, the value learning process with $\mathbb{T}^\pi$ is:

$$\mathbb{Q}_\theta^{\pi_i}(s, a, \chi_{\pi_i}) \leftarrow r + \gamma \mathbb{Q}_{\theta^-}^{\pi_i}\left(s', \pi_i(s'), \chi_{\pi_i}\right), \forall (s, a, r, s') \in \mathcal{B}. \tag{5}$$

Significantly different from previous work Faccio et al. (2020), $\mathcal{B}$ indicates the experience replay buffer shared by all policies. In other words, the value approximation of policy $\pi_i$ is not limited to the samples $\{(s, a, r, s', \pi_i(s'))\}$ generated by the interaction between the policy $\pi_i$ and the environment. The number of samples $\{(s, a, r, s', \pi_i(s'))\}$ is very limited and it's not really an off-policy evaluation manner using samples $\{(s, a, r, s', \pi_i(s'))\}$. Thus, GOPE greatly improves the sample efficiency of off-policy RL. Notably, the effectiveness of GOPE relies on policy representations, without which value generalization among policy space does not exist. That is, this learning manner, i.e., $\theta_t \xrightarrow{\mathcal{P}_{\pi \in \Pi_t^{GPI}}} \theta_t' \xrightarrow{\mathbb{T}_n^{\pi_{t+1}}} \theta_{t+1}$ alone may be ineffective under no-policy representations. Regarding this, we verified it in ablation experiments 5.2.

## 3.3 A Practical Implementation of the Proposed Algorithm

The Sec.3.1 and 3.2 detail how policy representation boosts the off-policy RL from both function approximation and off-policy learning. Next, combining the general and popular value estimation method, Clipped Double Q-learning (CDQ) adopted in off-policy RL Fujimoto et al. (2018); Haarnoja et al. (2018b), we propose a practical implementation of the proposed algorithm.

Following Double Q-learning algorithm Hasselt (2010); Van Hasselt et al. (2016), CDQ consists of double estimator, i.e., $Q_{\theta_1}, Q_{\theta_2}$. To alleviate the overestimation problem, CDQ proposes to simply upper-bound the less biased value estimate $Q_{\theta_2}$ by the biased estimate $Q_{\theta_1}$. Thus, for policy $\pi_\phi$, the update target that both critic $Q_{\theta_1}, Q_{\theta_2}$ share is $y = r + \gamma \min_{i=1,2} Q_{\theta_i^-}(s', \pi_\phi(s'))$. In this work, replacing $Q_{\theta_i}, Q_{\theta_i^-}$ with $\mathbb{Q}_{\theta_i}, \mathbb{Q}_{\theta_i^-}$, we propose a policy representation-based CDQ method. Furthermore, the value approximation of both critics is formulated below, for $i$=1, 2:

$$\mathbb{Q}_{\theta_i}(s, a, f_\psi(\pi_\phi)) \leftarrow r + \gamma \min_{i=1,2} \mathbb{Q}_{\theta_i^-}\left(s', \pi_\phi\left(s'\right), f_{\psi^-}(\pi_\phi)\right). \tag{6}$$

$f_\psi(\cdot)$ denotes the policy encoder with parameter $\psi$. In particular, in order to adapt to the target network $\mathbb{Q}_{\theta_i^-}$, we maintain a target policy encoder $f_{\psi^-}(\cdot)$. The target policy encoder $f_{\psi^-}(\cdot)$ is updated by $\psi^- \leftarrow \tau\psi + (1 - \tau)\psi^-$. Fig.7 illustrates the architecture of CDQ with policy representation. Additionally, we empirically investigate other architectures and perform experimental validation on four Mujoco-based environments. More results can be found in Appendix B. From Eq. 6, our algorithm only uses policy representations and is not limited to a particular off-policy RL algorithm, which greatly improves the generality of our algorithm. To illustrate the generality of our algorithm, we use CDQ-based TD3 Fujimoto et al. (2018) and SAC Haarnoja et al. (2018a) as the baselines respectively, and compare them with our algorithms **TD3-GOPE** and **SAC-GOPE**. Due to space limitations, we present only TD3-GOPE in Algorithm 1, and SAC-GOPE is presented in Appendix E.

---

**Algorithm 1** TD3-GOPE

---

**Initialize** critic networks $\mathbb{Q}_{\theta_1}, \mathbb{Q}_{\theta_2}$, actor network $\pi_\phi$ and policy encoder network $f_\psi$, with random parameters $\theta_1, \theta_2, \phi, \psi$, target networks $\theta_1^- \leftarrow \theta_1, \theta_2^- \leftarrow \theta_2, \phi^- \leftarrow \phi, \psi^- \leftarrow \psi$, replay buffer $\mathcal{B}, \mathcal{B}'$, policy buffer $\mathcal{D}$ and update interval $M$
for iteration $t = 0, 1, 2, \cdots$ do
    Select action $a$ and observe reward $r$ and new state $s'$. Store transition tuple $(s, a, r, s')$ in $\mathcal{B}$. Store policy $(\phi, \phi^-)$ in $\mathcal{D}$

> **Value learning of historical policies**
>
>     if iteration $t \% M = 0$ then
>         Sample mini-batch of $n$ policies from $\mathcal{D}$ and mini-batch of $N$ transitions $(s, a, r, s')$ from $\mathcal{B}$
>         for $v = 1, 2, \cdots, n$ do
>           $a' \leftarrow \pi_{\phi_v^-}(s')$, store $N$ transition tuple $(s, a, r, s', a', v)$ in $\mathcal{B}'$
>         end for
>         Sample mini-batch of $N$ transitions $(s, a, r, s', a', v)$ from $\mathcal{B}'$
>         Update critics and policy encoder parameter $\theta_i, \psi$ by approximating the action-value functions of historical policies with the experiences in $D, \mathcal{B}, \mathcal{B}'$ (Eq.6)
>         Update target networks $\theta_i^- \leftarrow \tau\theta_i + (1-\tau)\theta_i^-$, $\psi^- \leftarrow \tau\psi + (1-\tau)\psi^-$

> **Value learning of current policy**
>
>     Sample mini-batch of $N$ transitions $(s, a, r, s')$ from $\mathcal{B}$
>     Update critics and policy encoder parameter $\theta_i, \psi$ by approximating the action-value function of current policy with the experiences in $\mathcal{B}$ (Eq.6)
>     Update actor $\phi \leftarrow \operatorname{argmax}_\phi \mathbb{E}_{(s,a,r,s') \sim \mathcal{B}}(\mathbb{Q}_{\theta_1}(s, \pi_\phi(s), f_\psi(\pi_\phi)))$
>     Update target networks $\theta_i^- \leftarrow \tau\theta_i + (1-\tau)\theta_i^-$, $\phi^- \leftarrow \tau\phi + (1-\tau)\phi^-$, $\psi^- \leftarrow \tau\psi + (1-\tau)\psi^-$
> end for

---

## 4   Dynamic Masking for Policy Network Representation Learning

To derive general off-policy deep RL algorithms with GOPE, a tricky question is how to learn low-dimensional policy representations for better approximation and generalization of action-value function $\mathbb{Q}_\theta(\chi_\pi)$ across policies. Intuitively, policy representations optimized based on TD learning contain the most relevant features with value approximation. Thus, we consider using TD loss to optimize policy representations. For practical implementation, we further consider 1) the policy data sources for learning policy representations and 2) the encoder $f_\psi$ for learning policy representations.

### 4.1   Policy Data

In essence, any data characterizing policies can be used as the policy data sources, such as parameters of a policy network, trajectories, etc. However, compared with the trajectories with high randomness, the parameters of a policy network are usually available and highly deterministic. Thus, we learn policy representations from policy parameters. Generally, we consider a policy network to be an MLP with well-represented state features (e.g., features extracted by CNN for pixels or by LSTM for sequences) as input. In DRL, the size of policy networks is usually $64 \times 64$, $256 \times 256$, or larger. The high-dimensional policy parameters raise the question of whether to use all parameters to learn policy representations. This motivates a **hypothesis**: some neural nodes of the policy network may not be active in parameter updates and action decisions during the learning process, and the policy parameters associated with these neural nodes may not be helpful to learning policy representations. To verify the hypothesis, we investigate the evolvement of neural nodes of the policy network during the learning process. We first train a TD3 and a SAC agent on Mujoco-based tasks and store policies $\{\phi_i\}_{i=1,\cdots,N}$ at intervals of 200 steps over the course of training. Ranging from the initial policy $\pi_{\phi_1}$ to the final policy $\pi_{\phi_N}$, we iteratively accumulate the amount of parameter change of two adjacent policies, i.e., $\pi_{\phi_i}, \pi_{\phi_{i+1}}$ for each parameter dimension $k$, denoted as $\delta^k$. Assuming that the dimension

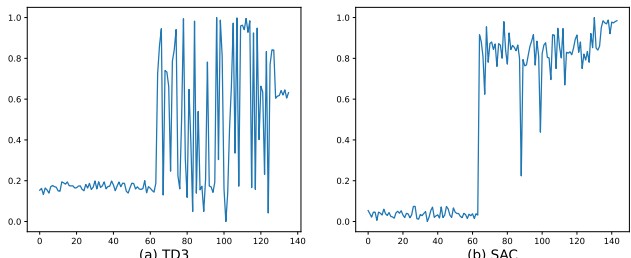

Figure 1: Average amount of parameter change of policy network neural nodes on the Ant during the learning process. The x-axis represents the indices of neural nodes, and the y-axis is the normalized average amount of parameter change.

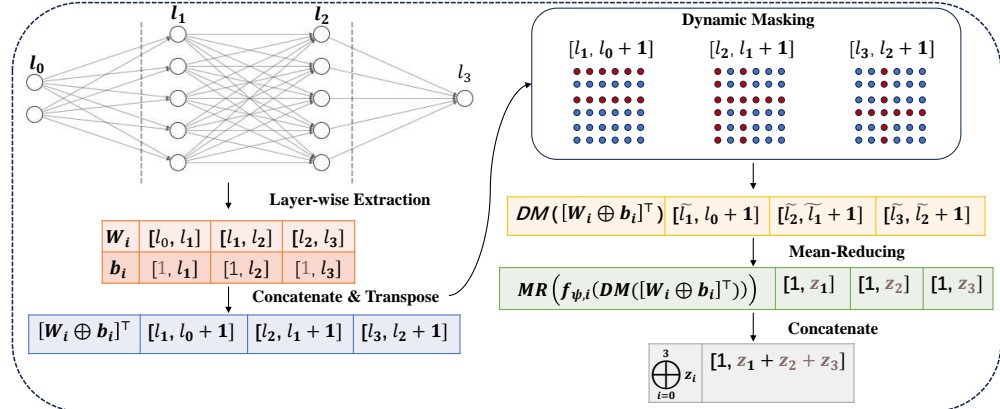

Figure 2: An illustration for Layer-wise Permutation-invariant Encoder with Dynamic Masking (LPE-DM). $l_0$, $l_{1,2}$ and $l_3$ denote the numbers of units for the input layer, hidden layer, and output layer, respectively. In dynamic masking, the red dots indicate the parameters being masked. Taking the first hidden node of $l_1$ as an example, its association parameters are the first row of $[l_1, l_0 + 1]$ and the first column of $[l_2, l_1 + 1]$.

of the policy parameters is $K$, then $\delta^k$ is defined as:

$$\delta^k = \sum_{i=1}^{i=N-1} |\phi_i^k - \phi_{i+1}^k|, k = 1 \cdots K, \tag{7}$$

where $\phi_i^k$ represents the parameter of the $k$th dimension for the policy $\pi_{\phi_i}$. Further, we calculate the average amount of parameter change in the association parameters for each neural node $j$ $(j = 1, \ldots, J)$ of the policy network, denoted as $\bar{\delta}_j$. In Fig. 1, we show the activity of neural nodes for the policy network on the Ant for both algorithms. The significant differences in the activity of neural nodes of the policy network, illustrated in Fig.1, provide empirical validation of our hypothesis.

## 4.2 POLICY ENCODING

Based on the empirical discovery above, a natural idea of learning policy representations is to only make use of the neural nodes that are active in parameter change. To this end, we propose *Dynamic Masking (DM)*: a method that allows us to dynamically mask neural nodes with low activity during the learning process. Given a policy $\pi_\phi$, it can be characterized using the policy parameters $\widetilde{\phi}_i$ associated with neural nodes with high activity. Specifically, in order to preserve the inter-layer structures of policy, $\widetilde{\phi}_i$ can be obtained by layer-wise masking nodes according to masking ratio $\eta$ and the amount of parameter change of nodes $\bar{\delta}_j$ of each layer. As the policy updates, we dynamically update the masked node set at an interval of 50 steps, and the policy representation is always calculated with the latest mask results.

To better characterize policies, we further propose a Layer-wise Permutation-invariant Encoder with Dynamic Masking (LPE-DM), $f_\psi(\widetilde{\phi})$. LPE-DM is illustrated in Fig. 2, which consists of five steps: Layer-wise Extraction, Concatenate & Transpose, Dynamic Masking, Mean-Reducing,

Table 1: Evaluation of TD3-GOPE in terms of learning stability and efficiency. The results of evaluation returns (± half a std) over 10 trials for algorithms are reported. The best results are bolded for each environment.

| Environment | Ave-Evaluation | | | Max-Evaluation | | |
|---|---|---|---|---|---|---|
| | *RanP* | *TD3* | *TD3-GOPE* | *RanP* | *TD3* | *TD3-GOPE* |
| HalfCheetah | -363.75±83.99 | 7866.79±546.25 | **9048.27±168.56** | -340.60±92.09 | 9920.17±750.17 | **11254.0±159.42** |
| Hopper | 21.43±8.23 | 2464.53±159.18 | **2778.23±101.82** | 22.72±8.73 | 3659.1±28.97 | **3666.16±26.83** |
| Walker2d | -7.76±1.67 | 2573.99±286.84 | **3241.44±194.73** | -6.99± 2.23 | 4187.83±287.79 | **4819.58±156.07** |
| Ant | 926.89±14.53 | 2265.99±97.2 | **3606.0±281.72** | 937.77±15.32 | 3474.56±219.35 | **5203.67±365.05** |
| InvDouPend | 72.53±5.79 | 8463.32±90.4 | **8761.04±48.12** | 78.34±9.45 | 9336.3±7.95 | **9355.73±1.39** |
| LunarLander | -206.35±68.91 | 226.08±9.61 | **241.40±6.03** | 146.22±41.30 | 292.26±1.91 | **292.58±2.57** |
| Norm. Agg. | 0 | 1 | **1.26 (↑ 26%)** | 0 | 1 | **1.16 (↑ 16%)** |

Table 2: Evaluation of SAC-GOPE in terms of learning stability and efficiency. The results of evaluation returns (± half a std) over 10 trials for algorithms are reported. The best results are bolded for each environment.

| Environment | Ave-Evaluation | | | Max-Evaluation | | |
|---|---|---|---|---|---|---|
| | *RanP* | *SAC* | *SAC-GOPE* | *RanP* | *SAC* | *SAC-GOPE* |
| HalfCheetah | -363.75±83.99 | 8765.53±133.77 | **9908.77±101.9** | -340.60±92.09 | 12431.71±52.78 | **13394.42±476.68** |
| Hopper | 21.43±8.23 | 2064.33±216.12 | **2180.57±98.04** | 22.72±8.73 | **3503.37±89.04** | 3295.36±117.82 |
| Walker2d | -7.76±1.67 | 3312.06±150.46 | **3748.67±189.07** | -6.99± 2.23 | 5102.93±213.06 | **5427.39±217.68** |
| Ant | 926.89±14.53 | 2416.79±169.40 | **3825.16±139.74** | 937.77±15.32 | 3911.47±525.99 | **5633.17±174.72** |
| InvDouPend | 72.53±5.79 | 8983.37±30.96 | **9019.36±31.56** | 78.34±9.45 | 9359.37±0.22 | **9359.53±0.26** |
| LunarLander | -206.35±68.91 | 182.97±13.08 | **232.61±11.14** | 146.22±41.30 | **284.0±1.35** | 283.23±1.75 |
| Norm. Agg. | 0 | 1 | **1.23 (↑ 23%)** | 0 | 1 | **1.11 (↑ 11%)** |

and Concatenate. Specifically, with respect to Mean-Reducing, $MR\left(f_{\psi,i}(DM([W_i \oplus b_i]^\top))\right) = \frac{1}{\bar{l}_i+1} \sum_{j=1}^{\bar{l}_i+1} f_{\psi,i}\left(DM(([W_i \oplus b_i]^\top))_{j,\cdot}\right)$. Each row of $DM([W_i \oplus b_i]^\top)$, indexing by the subscript $j, \cdot$, describes a transformation of the $i$-layer into the next layer. All the rows are fed into $f_{\psi,i}$ separately and are then averaged into $z_i$. The key to LPE-DM is to explicitly consider the characteristics of policy network structures (i.e., the intra-layer and inter-layer structures) and the variation of policy parameters during the learning process. Thus, it is an efficient encoding method tailored for policy network representation.

## 5 EXPERIMENTS

To evaluate our algorithm, we conduct experiments on the OpenAI Gym continuous control tasks Brockman et al. (2016). We run each task for 1 million time steps with evaluations every 5000 time steps, where each evaluation reports the average reward over 10 episodes with no exploration noise.

### 5.1 COMPARATIVE EVALUATION

In Sec.3.3, we present a practical implementation of our algorithm based on CDQ. To be specific, following the original CDQ, we also maintain a pair of critics i.e., $\mathbb{Q}_{\theta_1}(s, a, f_\psi(\pi_\phi))$, $\mathbb{Q}_{\theta_2}(s, a, f_\psi(\pi_\phi))$ along with a single actor $\pi_\phi$. For our implementation for the critic, we utilize a two-layer feedforward neural network of 256 and 256 hidden nodes, respectively. We optimize a two-layer policy network with 64 hidden nodes for each layer, resulting in over 4k to 10k policy parameters depending on the tasks. The policy storage interval is 10 update steps and we maintain a proximal policy buffer $D$ of size 2000, which maintains a low memory overhead. The above setting applies to all experiments in the paper. We list common hyperparameters in Table 5 and Table 6.

The experimental results of our implementations (**TD3-GOPE**, **SAC-GOPE**) and the corresponding baselines (**RA** (Random Agent), **TD3** and **SAC**) are reported in Table 1 and Table 2, respectively. We defer the full learning curves to Appendix F (see Fig. 8, 9). In Table 1 and Table 2, we report two performance metrics: 1) the average performance attained over the course of training (denoted **Ave-Evaluation**), which is a measure of the stability of RL algorithms over the course of training and 2) the max performance attained by the algorithm after a fixed number of gradient steps (denoted **Max-Evaluation**). In addition, we evaluate the aggregated improvement (denoted **Norm. Agg.**) of our algorithm on multiple tasks using a random agent and TD3, SAC

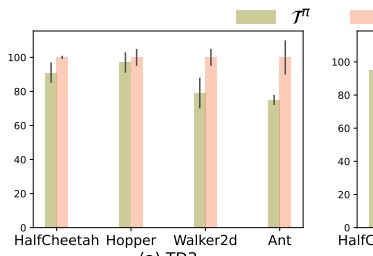
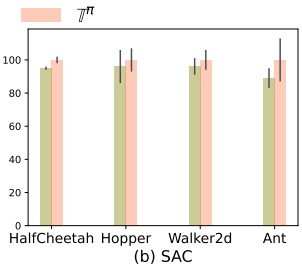

for normalization, respectively. The empirical results demonstrate that our methods (**TD3-GOPE**, **SAC-GOPE**) consistently improve the benchmarks on all tasks tested. Moreover, compared with TD3, our method improves **26%** and **16%** with respect to **Ave-Evaluation** and **Max-Evaluation**, respectively; Compared with SAC, our method improves **23%** and **11%** with respect to **Ave-**

Figure 3: The $x$-axis represents four tasks, and the $y$-axis is the normalized average return. Our method ($\mathbb{T}^\pi$) in each environment is chosen as a normalized baseline. **Conclusion:** In both TD3 and SAC cases, our method outperforms the original Bellman operator ($\mathcal{T}^\pi$), which empirically proves the value generalization with $\mathbb{T}^\pi$.

**Evaluation** and **Max-Evaluation**, respectively. The significant improvement demonstrates the advantages of our algorithm in terms of learning efficiency and stability.

## 5.2 ABLATION STUDY

Next, we investigate further the efficacy of each component of the proposed method. Concretely, we conduct ablation experiments to answer the following three questions:

**1.** *Can the implicit generalization of Bellman operator $\mathbb{T}^\pi$ with policy representation offer better function approximation?* (Sec.3.1) **2.** *Can generalized off-policy evaluation further improve value generalization and learning efficiency?* (Sec.3.2) **3.** *Is LPE-DM an effective method to encode policy network parameters?* (Sec.4)

To answer the first question, we compare the performance difference between $\mathcal{T}^\pi$ and $\mathbb{T}^\pi$. We use TD3 and SAC as practical implementations of $\mathcal{T}^\pi$, respectively. Correspondingly, we replace only $Q_\theta$ with $\mathbb{Q}_\theta(\chi_\pi)$ in the TD3 and SAC as practical implementations of $\mathbb{T}^\pi$ with policy representation, respectively. The experimental results in Fig.3 show that the Bellman operator $\mathbb{T}^\pi$ outperforms its original counterpart, which empirically demonstrates that implicit generalization of Bellman operator $\mathbb{T}^\pi$ with policy representation offer better function approximation.

To answer the second question, we propose two groups of ablation experiments. For brevity, we abbreviate the policy representation as PR. The first one is the proposed GOPE *vs.* GOPE-w/o PR. We retain the generalized off-policy evaluation manner but do not use the policy representation, named GOPE-w/o PR. On the contrary, the other is GOPE *vs.* OPE-w/PR. We retain the policy representation but do not use the generalized off-policy evaluation manner, named OPE-w/PR, for which the learning frequency of the action-value function is identical to GOPE. Besides, we use TD3 and SAC as practical implementations of OPE. If the performance can be improved by learning $Q_\theta$ in the generalized off-policy evaluation manner or by learning $\mathbb{Q}_\theta(\chi_\pi)$ more times, then GOPE-w/o PR and OPE-w/PR should be comparable to our algorithm. Instead, Fig. 4 shows that the performance of GOPE-w/o PR and OPE-w/PR is much lower than the GOPE. The empirical results show that the generalized off-policy evaluation with policy representation is crucial to further improve the performance of off-policy RL algorithm and the two (i.e., GOPE and PR) are complementary.

To answer the third question, we consider five policy representation learning methods in terms of both policy encoder and dynamic masking methods. To highlight the superiority of LPE, we compared it with unencoded policy parameters (Params) Faccio et al. (2020) and a multilayer perceptron (MLP)-based policy encoder. The policy parameters, as the source data for policy representations, can themselves be used as an uncompressed policy representation. The MLP-based policy encoder flattens the policy into a vector input and utilizes a two-layer feedforward neural network of 256 and 256 hidden nodes. In addition, we also compare the fixed randomly generated policy representation of 64 dimensions, named as RPR. LPE-DM represents our method. Fig. 5 reports the experimental results on different methods respectively. Obviously, as an original policy representation, the policy parameters are far worse than other policy encoding methods, mainly due to their high-dimensional and highly nonlinear, offering no help in the function approximation. Compared to the MLP-based policy encoder, the most significant characteristic of LPE is that it explicitly considers both the intra-layer and inter-layer structures. LPE-DM shows consistent advantages in the tested tasks, especially in Ant environment.

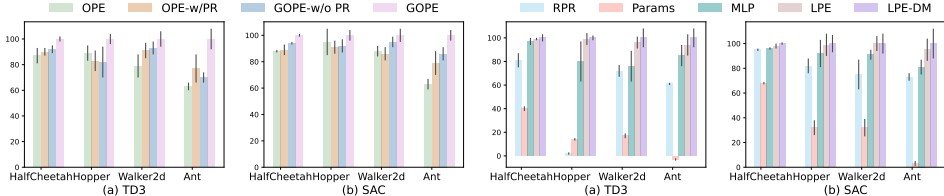

Figure 4: The $x$-axis represents four test tasks, and the $y$-axis is the normalized average re-turn. Our method (GOPE) in each environ-ment is chosen as a normalized baseline. **Con-clusion:** In both TD3 and SAC cases, our method is better than the comparison meth-ods (OPE, OPE-w/PR, GOPE-w/o PR), which empirically proves the efficacy of generalized off-policy evaluation with policy representa-tion.

Figure 5: The $x$-axis represents four test tasks, and the $y$-axis is the normalized average re-turn. Our method (LPE-DM) in each task is chosen as a normalized baseline. **Conclusion:** In both TD3 and SAC cases, our method is better than the comparison methods (RPR, Params, MLP, LPE), which empirically illus-trates that LPE with DM is an efficacy method for learning policy representations from pol-icy parameters.

## 5.3 ADDITIONAL DISCUSSION

In this section, we discuss the correlation between value generalization and learning performance. We first define a *value generalization* metric, $\Delta$ based on TD update rule:

$$\Delta = \sum_{t=1}^{T} \mathbb{E}_{(s,a,r,s')\sim\mathcal{B}}[\epsilon_{\pi_t} - \epsilon_{\pi_{t+1}}], \tag{8}$$

$$\epsilon_{\pi_t} = \sum_{i=1}^{i=2}(\mathbb{Q}_{\theta_i}(s, a, \chi_{\pi_t}) - y)^2, \epsilon_{\pi_{t+1}} = \sum_{i=1}^{i=2}(\mathbb{Q}_{\theta_i}\left(s, a, \chi_{\pi_{t+1}}\right) - y)^2, \tag{9}$$

where $y = r + \gamma \min_{i=1,2} \mathbb{Q}_{\theta_i^-}\left(s', \pi_{t+1}\left(s'\right), \chi_{\pi_{t+1}}\right)$. $\epsilon_{\pi_t} - \epsilon_{\pi_{t+1}}$ measures the difference of fitting TD target $y$ using $\mathbb{Q}_\theta(\chi_\pi)$ with two adjacent policy representations (i.e., $\chi_{\pi_t}, \chi_{\pi_t+1}$, marked in blue). We repeat the experiment 30 trials with different random seeds on HalfCheetah and Walker2d tasks, respectively, and store the evaluation returns and the value generalization metric of each experiment. To increase the reliability of the experimental results, we adopt two correlation coefficients, *Pearson (P-Corr)* and *Spearman (S-Corr)*. From Fig. 6, we can obtain 1) value generalization $\Delta$ is positive ($x$-axis) (i.e., on the whole, $\epsilon_{\pi_t} \geq \epsilon_{\pi_{t+1}}$), which indicates that $\mathbb{Q}_\theta(\chi_\pi)$ has the ability of positive generalization. 2) The results of the two correlation coefficients are around 0.6, which indicates that the positive generalization of $\mathbb{Q}_\theta(\chi_\pi)$ improves learning performance.

## 6 CONCLUSION

This paper proposes a sim-ple, efficient, and generalized deep off-policy RL algorithm, GOPE. It adopts the pro-posed Bellman operator $\mathbb{T}^\pi$ and a generalized off-policy evaluation manner with low-dimensional policy representa-tion for better value approxi-mation and value generaliza-tion and thus greatly improves sample efficiency in off-policy

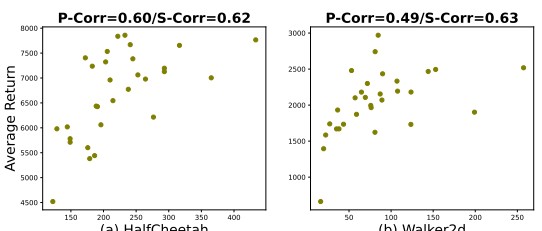

Figure 6: The correlation between the value generalization ($x$-axis) and the average return ($y$-axis) of our algorithm. Each dot represents a random trial. **Conclusion:** *Pearson* and *Spearman* both show a weak positive correlation between the above two.

RL. Furthermore, this work first investigates the evolvement of neural nodes of the policy network during the learning process and proposes a policy representation learning approach towards policy networks. The empirical results demonstrate that our algorithm is general enough to incorporate into other off-policy algorithms.

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

## A  PROOF OF THEOREM 3.3

*Proof.* Due to *Triangle Inequality*, we have:

$$f_{\hat{\theta}_t}(\pi_t) + \|\mathbb{Q}_{\hat{\theta}_t}(\chi_{\pi_t}) - Q^{\pi_{t+1}}\| \geq \|Q^{\pi_t} - Q^{\pi_{t+1}}\|$$
$$i.e., \|\mathbb{Q}_{\hat{\theta}_t}(\chi_{\pi_t}) - Q^{\pi_t}\| + \|\mathbb{Q}_{\hat{\theta}_t}(\chi_{\pi_t}) - Q^{\pi_{t+1}}\| \geq \|Q^{\pi_t} - Q^{\pi_{t+1}}\|$$

(10)

Combined with condition

$$f_{\hat{\theta}_t}(\pi_t) + f_{\hat{\theta}_t}(\pi_{t+1}) \leq \|Q^{\pi_t} - Q^{\pi_{t+1}}\|$$
$$i.e., \|\mathbb{Q}_{\hat{\theta}_t}(\chi_{\pi_t}) - Q^{\pi_t}\| + \|\mathbb{Q}_{\hat{\theta}_t}(\chi_{\pi_{t+1}}) - Q^{\pi_{t+1}}\| \leq \|Q^{\pi_t} - Q^{\pi_{t+1}}\|$$

(11)

Chain above two inequality,

$$f_{\hat{\theta}_t}(\pi_t) + f_{\hat{\theta}_t}(\pi_{t+1}) \leq \|Q^{\pi_t} - Q^{\pi_{t+1}}\| \leq f_{\hat{\theta}_t}(\pi_t) + \|\mathbb{Q}_{\hat{\theta}_t}(\chi_{\pi_t}) - Q^{\pi_{t+1}}\|$$
$$i.e., \|\mathbb{Q}_{\hat{\theta}_t}(\chi_{\pi_{t+1}}) - Q^{\pi_{t+1}}\| \leq \|\mathbb{Q}_{\hat{\theta}_t}(\chi_{\pi_t}) - Q^{\pi_{t+1}}\|$$

(12)

$\square$

## B  DETAILS OF THE POLICY REPRESENTATION-BASED CLIPPED DOUBLE Q-LEARNING METHOD

In this work, we discuss four variants for policy representation-based CDQ implementation. Concretely, we consider the TD3 algorithm based on CDQ which maintains a pair of actors $(\pi_\phi, \pi_{\phi^-})$ and critics $(Q_{\theta_1}, Q_{\theta_2})$. In the first variant, we maintain a pair of policy encoders $(f_\psi(\phi), f_{\psi^-}(\phi))$, taking the parameters $\phi$ of policy $\pi_\phi$ as input. The value approximation of both critics is formulated as: ① $\mathbb{Q}_{\theta_i}(s, a, f_\psi(\phi)) \leftarrow r + \gamma \min_{i=1,2} \mathbb{Q}_{\theta_i^-}(s', \pi_{\phi^-}(s'), f_{\psi^-}(\phi))$. In the second variant, we maintain a pair of policy encoders $(f_\psi(\phi), f_{\psi^-}(\phi^-))$, taking the parameters $\phi$ of policy $\pi_\phi$ and the parameters $\phi^-$ of policy $\pi_{\phi^-}$ as input, respectively. The value approximation of both critics is formulated as: ② $\mathbb{Q}_{\theta_i}(s, a, f_\psi(\phi)) \leftarrow r + \gamma \min_{i=1,2} \mathbb{Q}_{\theta_i^-}(s', \pi_{\phi^-}(s'), f_{\psi^-}(\phi^-))$. In the third variant, we maintain a pair of policy encoders $(f_\psi(\phi^-), f_{\psi^-}(\phi^-))$, taking the parameters $\phi^-$ of policy $\pi_{\phi^-}$ as input. The value approximation of both critics is formulated as: ③ $\mathbb{Q}_{\theta_i}(s, a, f_\psi(\phi^-)) \leftarrow r + \gamma \min_{i=1,2} \mathbb{Q}_{\theta_i^-}(s', \pi_{\phi^-}(s'), f_{\psi^-}(\phi^-))$. In the fourth variant, we maintain two pairs of policy encoders $(f_{\psi_{1,2}}(\phi), f_{\psi_{1,2}^-}(\phi))$, taking the parameters $\phi$ of policy $\pi_\phi$ as input. The value approximation of both critics is formulated as: ④ $\mathbb{Q}_{\theta_i}(s, a, f_{\psi_i}(\phi)) \leftarrow r + \gamma \min_{i=1,2} \mathbb{Q}_{\theta_i^-}(s', \pi_{\phi^-}(s'), f_{\psi_i^-}(\phi))$. Table B reports the experimental results of the four variants (PRCDQ-v1, PRCDQ-v2, PRCDQ-v3, PRCDQ-v4) as well as the baselines (RA, TD3). The empirical results show the superiority of the first variant which is adopted in our algorithm.

## C  DETAILS OF GENERALIZED OFF-POLICY EVALUATION WITH POLICY REPRESENTATION

Policy representations endow $\mathbb{Q}_\theta(\chi_\pi)$ with the property of generalizing across policies. However, during the learning process, the knowledge obtained through the value learning of early historical policies may be too old to benefit the generalization of $\mathbb{Q}_\theta(\chi_\pi)$ across policies. Therefore, we propose to only perform the value learning of the proximal policies of the current policy. To be specific, the policy storage interval is 10 update steps and we maintain a proximal policy buffer $D$ of size 2000, which maintains a low memory overhead. With TD3 as the baseline, Table C reports the experimental results of different buffer sizes. Compared with the cases of $size = 0$ (i.e., value learning without performing historical policies) and $size = all$ (i.e., value learning with performing all historical policies), the value learning of proximal historical policies ($size = 500$, $size = 2000$) obtains better results with respect to the Ave-Evaluation, which demonstrates that the advantages of the proposed GOPE in terms of stability and learning efficiency. In the future, we will explore better historical policy sampling methods.

Table 3: Discussion of different implementation variants for policy representation-based Clipped Double Q-learning. Average evaluation returns and Max evaluation returns (± half a std) over 10 trials for algorithms. The best results are bolded.

| Environment | Ave-Evaluation | | | | | |
|---|---|---|---|---|---|---|
| | *RanP* | *TD3* | *PRCDQ-v1* | *PRCDQ-v2* | *PRCDQ-v3* | *PRCDQ-v4* |
| HalfCheetah | -363.75±83.99 | 7866.79±546.25 | 8620.37±110.55 | 8181.73±405.1 | 7961.49±310.02 | 8724.56±138.04 |
| Hopper | 21.43±8.23 | 2464.53±159.18 | 2544.11±122.45 | 2588.21±129.02 | 2427.17±38.0 | 2536.36±131.46 |
| Walker2d | -7.76±1.67 | 2573.99±286.84 | 3271.57±169.02 | 3147.0±259.95 | 3302.85±152.21 | 3425.49±86.56 |
| Ant | 926.89±14.53 | 2265.99±97.2 | 3034.7±300.83 | 2998.42±396.9 | 2645.82±204.38 | 2497.67±72.48 |
| Norm. Agg. | 0 | 1 | **1.24 (↑ 24%)** | 1.21 (↑ 21%) | 1.14 (↑ 14%) | 1.16 (↑ 16%) |
| Environment | Max-Evaluation | | | | | |
| | *RanP* | TD3 | *PRCDQ-v1* | *PRCDQ-v2* | *PRCDQ-v3* | *PRCDQ-v4* |
| HalfCheetah | -340.6±92.09 | 9920.17±750.17 | 10664.98±198.68 | 10637.38±271.78 | 10461.47±530.02 | 11182.97±236.79 |
| Hopper | 22.72±8.73 | 3659.1±28.97 | 3644.91±29.2 | 3633.78±36.7 | 3637.92±49.27 | 3637.23±35.15 |
| Walker2d | -6.99±2.23 | 4187.83±287.79 | 4854.9±226.1 | 4951.48±229.64 | 4835.86±228.55 | 5196.72±159.75 |
| Ant | 937.77±15.32 | 3474.56±219.35 | 4609.4±549.87 | 4479.09±436.54 | 4039.76±519.72 | 3801.51±412.43 |
| Norm. Agg. | 0 | 1 | **1.17 (↑ 17%)** | 1.16 (↑ 16%) | 1.11 (↑ 11%) | 1.12 (↑ 12%) |

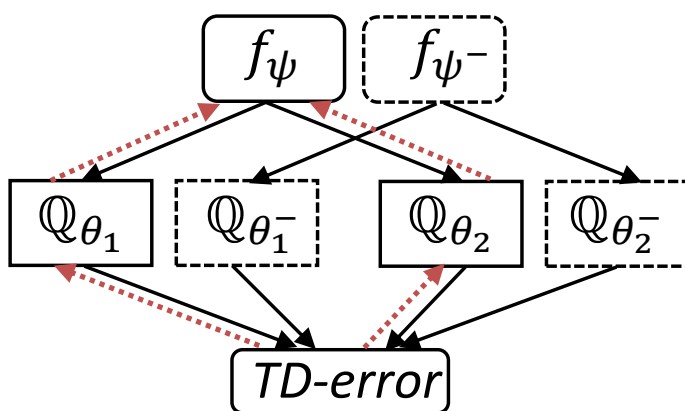

Figure 7: The illustration diagram for the policy representation-based Clipped Double Q-learning. The black solid arrows indicate forward propagation and the red dashed arrows are backward propagation.

## D  HYPERPARAMETERS

Table 5 shows the common hyperparameters of the algorithm used in all our experiments. Table 6 shows the structure of the actor network and critic network for TD3-GOPE and SAC-GOPE. To be specific, we utilize a two-layer feed-forward neural network of 64 and 64 hidden units with ReLU activation (except for the output layer) for the actor network. Similarly, the critic network also uses a two-layer feed-forward neural network of 256 and 256 hidden units with ReLU activation.

## E  PSEUDO-CODE OF SAC-GOPE

The pseudo-code of the proposed algorithm, SAC-GOPE is in Algorithm 2.

## F  COMPLETE LEARNING CURVES

Fig. 8 and Fig. 9 shows the learning curves of RanP, TD3, SAC, TD3-GOPE and SAC-GOPE corresponding to the results in Table 1 and Table 2.

Table 4: Discussion of buffer size for proximal policies. Average evaluation returns and Max evaluation returns (± half a std) over 10 trials for algorithms. The best results are bolded for each environment.

| Environment | Ave-Evaluation | | | |
|---|---|---|---|---|
| | *size=all* | *size=2000* | *size=500* | *size=0* |
| HalfCheetah | 8990.75±133.98 | **9048.27±168.56** | 8920.27±223.68 | 8693.77±226.57 |
| Hopper | 2758.75±108.01 | 2778.23±101.82 | **2779.54±86.74** | 2579.82±38.96 |
| Walker2d | 2268.95±371.82 | 3241.44±194.73 | **3496.18±217.25** | 3416.55±258.93 |
| Ant | 2579.88±717.87 | **3606.0±281.72** | 3073.98±375.69 | 3223.43±245.85 |
| Environment | Max-Evaluation | | | |
| | *size=all* | *size=2000* | *size=500* | *size=0* |
| HalfCheetah | 11152.91±124.74 | **11254.0±159.42** | 11110.28±118.06 | 10803.12±447.49 |
| Hopper | 3657.07±19.92 | 3666.16±26.83 | 3707.71±25.66 | **3728.67±40.22** |
| Walker2d | 3980.16±74.25 | 4819.58±156.07 | 4881.04±209.69 | **5123.22±263.97** |
| Ant | 4253.95±526.31 | 5203.67±365.05 | 4424.04±556.08 | **5288.47±323.18** |

---

**Algorithm 2** SAC-GOPE

---

**Initialize** critic networks $\mathbb{Q}_{\theta_1}$, $\mathbb{Q}_{\theta_2}$, actor network $\pi_\phi$ and policy encoder network $f_\psi$, with random parameters $\theta_1$, $\theta_2$, $\phi$, $\psi$, target networks $\theta_1^- \leftarrow \theta_1$, $\theta_2^- \leftarrow \theta_2$, $\psi^- \leftarrow \psi$, replay buffer $\mathcal{B}, \mathcal{B}'$, policy buffer $\mathcal{D}$ and update interval $M$
for iteration $t = 0, 1, 2, \cdots$ do
    Select action $a$ and observe reward $r$ and new state $s'$. Store transition tuple $(s, a, r, s')$ in $\mathcal{B}$. Store policy $\phi$ in $\mathcal{D}$

> **Value learning of historical policies**
>
> if iteration $t \% M = 0$ then
>     Sample mini-batch of $n$ policies from $\mathcal{D}$ and mini-batch of $N$ transitions $(s, a, r, s')$ from $\mathcal{B}$
>     for $v = 1, 2, \cdots, n$ do
>         $a', \log \pi_{\phi_v}(\cdot|s') \leftarrow \pi_{\phi_v}(s')$, store $N$ transition tuple $(s, a, r, s', a', \log \pi_{\phi_v}(\cdot|s'), v)$ in $\mathcal{B}'$
>     end for
>     Sample mini-batch of $N$ transitions $(s, a, r, s', a', \log \pi_{\phi_v}(\cdot|s'), v)$ from $\mathcal{B}'$
>     $y \leftarrow r + \gamma(\min_{i=1,2} \mathbb{Q}_{\theta_i^-}(s', a', f_{\psi^-}(\pi_{\phi_v})) - \alpha \log \pi_{\phi_v}(\cdot|s'))$
>     Update critics and policy encoder parameter
>     $\theta_i, \psi \leftarrow \arg\min_{\theta_i, \psi} \mathbb{E}_{(s,a,r,s',a',\log \pi_{\phi_v}(\cdot|s'),v)\sim\mathcal{B}'} \sum_{i=1}^{i=2} \frac{1}{2}(y - \mathbb{Q}_{\theta_i}(s, a, f_\psi(\pi_{\phi_v})))^2$
>     Update target networks $\theta_i' \leftarrow \tau\theta_i + (1-\tau)\theta_i'$, $\psi' \leftarrow \tau\psi + (1-\tau)\psi'$

> **Value learning of current policy**
>
> Sample mini-batch of $N$ transitions $(s, a, r, s')$ from $\mathcal{B}$
> $y \leftarrow r + \gamma(\min_{i=1,2} \mathbb{Q}_{\theta_i^-}(s', \pi_\phi(s'), f_{\psi^-}(\pi_\phi)) - \alpha \log \pi_\phi(\cdot|s'))$
> Update critics and policy encoder parameter
> $\theta_i, \psi \leftarrow \arg\min_{\theta_i, \psi} \mathbb{E}_{(s,a,r,s')\sim\mathcal{B}} \sum_{i=1}^{i=2} \frac{1}{2}(y - \mathbb{Q}_{\theta_i}(s, a, f_\psi(\pi_\phi)))^2$
> Update actor $\phi \leftarrow \arg\max_\phi \mathbb{E}_{(s,a,r,s')\sim\mathcal{B}}(\min_{i=1,2} \mathbb{Q}_{\theta_i}(s, \pi_\phi(s), f_\psi(\pi_\phi)) - \alpha \log \pi_\phi(\cdot|s))$
> Update target networks $\theta_i^- \leftarrow \tau\theta_i + (1-\tau)\theta_i^-$, $\psi^- \leftarrow \tau\psi + (1-\tau)\psi^-$
> end for

Table 5: Common hyperparameters. We use '-' to denote the 'not applicable' situation.

| Hyperparameters | TD3-GOPE | SAC-GOPE |
|---|---|---|
| Actor Learning Rate | $10^{-3}$ | $3\times10^{-4}$ |
| Critic Learning Rate | $10^{-3}$ | $3\times10^{-4}$ |
| Target Action Noise | 0.2 | - |
| Actor Training Interval | 2 steps | 1 step |
| Masking Ratio ($\eta$) | 0.6 | 0.6 |
| Discount Factor ($\gamma$) | 0.99 | 0.99 |
| Soft Replacement Ratio | 0.005 | 0.002 |
| Replay Buffer Size | 200k time steps | 200k time steps |
| Batch Size | 100 | 128 |
| Training Interval | 1 step | 1 step |
| Optimizer | Adam | Adam |
| Masking Ratio ($\eta$) | 0.6 | 0.6 |
| Policy Representation Dimension (*pr dim*) | 64 | 64 |
| Update interval ($M$) | 10 | 10 |

Table 6: Structure of actor network and critic network. We use '-' to denote the 'not applicable' situation.

| Method | Layer | Actor Network ($\pi(a|s)$) | Critic Network ($\mathbb{Q}(\chi_\pi)$) |
|---|---|---|---|
| TD3-GOPE | Fully Connected | (state dim, 64) | (state dim + action dim + pr dim, 256) |
| | Activation | ReLU | ReLU |
| | Fully Connected | (64, 64) | (256, 256) |
| | Activation | ReLU | ReLU |
| | Fully Connected | (64, action dim) | (256, 1) |
| | Activation | tanh | None |
| SAC-GOPE | Fully Connected | (state dim, 64) | (state dim + action dim + pr dim, 256) |
| | Activation | ReLU | ReLU |
| | Fully Connected | (64, 64) | (256, 256) |
| | Activation | ReLU | ReLU |
| | Fully Connected | (64, action dim) | (256, 1) |
| | Activation | None | None |
| | Fully Connected | (64, action dim) | - |
| | Activation | None | - |

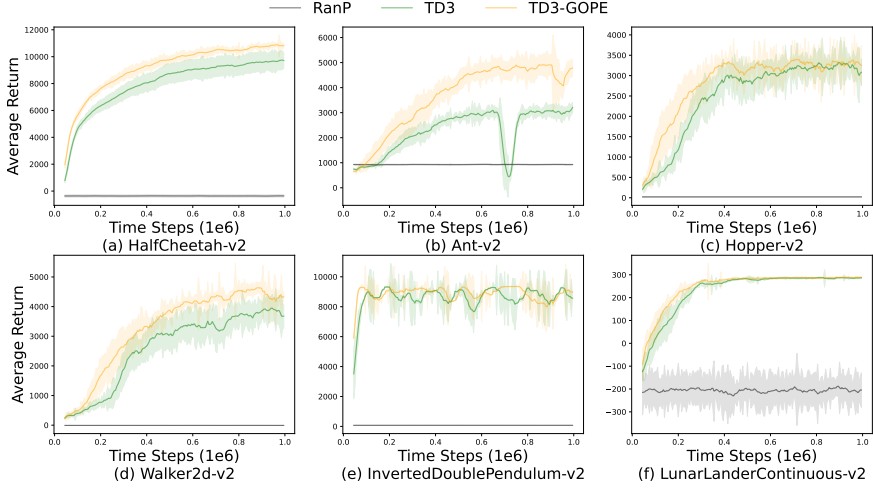

Figure 8: Learning curves for the OpenAI gym continuous control tasks. The shaded region represents half a standard deviation of the average evaluation over 10 trials. Curves are smoothed uniformly for visual clarity.

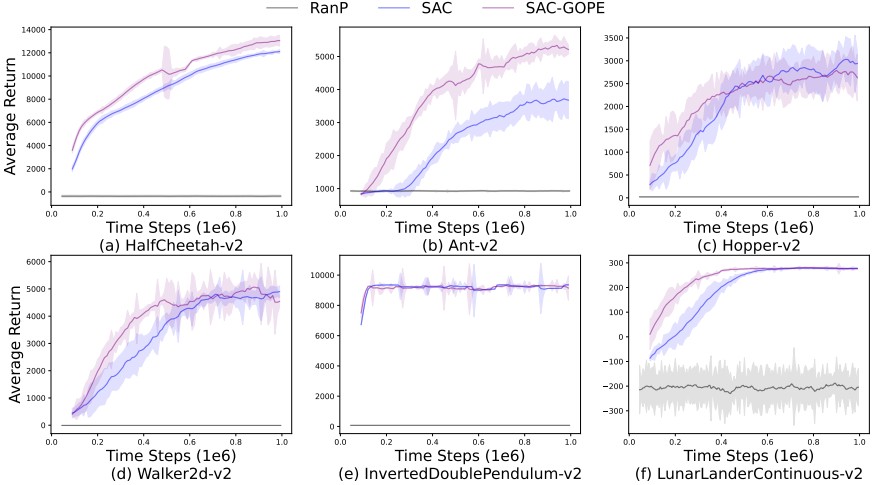

Figure 9: Learning curves for the OpenAI gym continuous control tasks. The shaded region represents half a standard deviation of the average evaluation over 10 trials. Curves are smoothed uniformly for visual clarity.

