# OpenReview forum: "Improving Sample Efficiency in Off-policy RL with Low-dimensional Policy Representation"
_ICLR.cc/2024/Conference — Submitted to ICLR 2024_

### Official Review · Reviewer_QhXs · 2023-10-30

**Soundness:** 2 fair
**Presentation:** 1 poor
**Contribution:** 2 fair
**Rating:** 3
**Confidence:** 3

**Summary:**

This paper studies the use of policy representations to condition Q-functions for better off-policy RL. The paper proposes an algorithm and practical details to condition a Q-function in this way for continuous control deep RL tasks.

**Strengths:**

The proposed algorithm seems simple and conceptually easy to grasp, and the reported performance improvement over baseline seems decent.

**Weaknesses:**

This paper has some issues that need resolving.

My biggest issue is that I'm not sure why it's reasonable to expect conditioning a Q-function on a representation of the policy's current parameters would naturally lead to superior performance. It seems like Definition 3.2 is supposed to fill that role, but I think you could write the same definition for any arbitrary conditioning factor. I'm willing to believe that the policy representation helps, but I don't understand why it should as either a pre-facto/theoretical or post-facto/experimental argument.

Secondly, the writing in the paper seems rather vague and poorly-worded in many places. In particular, the abstract and introduction both tell me almost nothing about the actual method, and there's a lot of confusing wording throughout. I suspect that resolving this issue will also resolve a lot of the specific issues I highlighted in the Questions section, but it somewhat dominates because it's hard to follow the arguments being made at times.

Ultimately, I think the method as I understand it is interesting and reasonable, but I don't follow the motivation for it very well, and the experimental evaluation seems somewhat limited (or at least it's not clear to me why this is a sufficient set of experiments to validate the method). With some heavy revision I think this could be a good paper, and I'm not sure it needs a ton of extra experiments or analysis (some more comparisons, perhaps?), but as is this paper needs more time in the oven.

**Questions:**

-The title advertising sample efficiency seems a bit misleading. I don't see any variation in the experiments on number of timesteps, and the topic of sample efficiency relative to the baseline doesn't seem to be a focus.

-The abstract seems somewhat vague and low information. It tells me that this paper is using a policy representation to improve off-policy RL, but doesn't tell me much more than that. I'd suggest re-writing it to add more details about the method.

-Likewise, the introduction doesn't tell me very much about the paper, or even how it differs from the previous work discussed, other than to tell me that it is. I'd suggest doing more in this section to motivate your method and explain how it differs from other methods using policy representations.

-I don't think this paper actually describes what a policy representation is in general anywhere other than the brief definition in Section 3.1. I'd recommend adding a section (maybe in Section 2?) describing what a policy representation is and how they have been used by previous works.

-Definition 3.2 makes no sense to me as written- It should be impossible to approximate a state conditioned function Q(s,a) using a function conditioned solely on a representation of a policy such as Q(X_\pi). Even if I assume the intent was a representation-conditioned function Q(s, a, X_\pi), I'm not seeing anything that motivates X_\pi as a particularly useful thing to condition on.

-While theorem 3.3 seems to be correct, I don't see any reason to believe that f_\theta(\pi) should be particularly small, i.e. I'm not sure the condition f(\pi_t) + f(\pi_{t+1}) <= ||Q^{\pi_t} - Q^{\pi_{t+1}}|| is likely to hold in practice, given the above point about a function not conditioned on state being a poor approximator of a state-conditional one.

-The metric "ave-evaluation" is confusing. Is this the average per-trial score across all trials? This seems like an odd and uninformative metric since there's a lot of different training curves that could produce the same value.

-There's a bunch of prior work cited on policy representations, but none of it was compared to. Is it truly the case that nobody has thought to condition a Q-function on a representation of its policy before?

-How is Figure 3 an ablation, isn't this just testing versus the baseline? It's not clear what the two different Bellman operators are if not the full GOPE method versus the baseline.

-Minor gripe, the spacing between Figures 4 and 5 is too small- they look like one figure, and trying to read their captions is confusing due to the lack of spacing.

-The difference between LPE and LPE-DM in Figure 5 looks to be well within error. Perhaps it would be simpler to drop the dynamic masking component given it doesn't seem to affect performance much?

-I'm not sure what Figure 6 is doing, or why this value generalization metric is relevant to plot versus average return (across an entire training run?)

---

### Official Review · Reviewer_WhXc · 2023-10-30

**Soundness:** 2 fair
**Presentation:** 1 poor
**Contribution:** 2 fair
**Rating:** 3
**Confidence:** 4

**Summary:**

The paper proposes to augment a deep RL agent's representation with the encoded policy. The paper considers methods that learn action-value functions (SAC, TD3) in combination with the policy representation LPE-DM. The two main claims are that policy representations help improve performance relative to only the usual state representations, and that the methods that use policy representations are stronger off-policy methods.

The evaluation shows that the policy representation improves the performance in control tasks relative to the SAC and TD3 baselines. The paper also ablates different components of the proposed method, namely, the policy representation and (if I understood correctly) the use of all the experience generated during training for performing policy updates.

The paper also investigates the relationship between policy representations, the quality of function approximation and performance, with a combination of theoretical and empirical results.

Overall, while I feel positive about the results in the paper, I had trouble understanding the main idea of the paper (LPE-DM) in detail, and I think that lack of clarity subtracts from the strength of the support for the main claims.

I think increasing the clarity of the paper in different sections would improve it, and I would increase the score for a clearer paper that provided a more detailed description of LPE-DM.

### Score rationale

Presentation: Unfortunately I had trouble getting a clear understanding of the main method, the paper has some technical issues and some sections are harder to read through. I am open to revising this score up if the presentation improves.

Soundness and Contribution: The lack of clarity on the main method makes it difficult for me to give higher ratings for soundness and contribution. I would be happy to revise these two up to "good" if I could understand LPE-DM better. I am also open revise the two scores further up if I get a better understanding of the impact and significance of the improvements.

**Strengths:**

I find the premise of the paper, the claims and the results interesting. The ablation study is interesting, informative and well executed.

**Weaknesses:**

Unfortunately, I found the paper difficult to understand, with technical issues and not enough clarity for me to feel confident that I could assess or reproduce the main idea.

**Questions:**

Thank you for your submission. I think the ideas and results in this paper are interesting, and I would like to be able to understand them, so I encourage you to clarify the presentation. This will make the method easier to reproduce, adopt and build on. I have some suggestions for improvement below that I hope you will find helpful.

The method being proposed should be very clear. There should be more detail and clarity about how the policy encoding is happening. Section 4.2 seems to be part of the core innovation in the paper (how the policy is represented), so improving its clarity will improve the paper. Also please provide enough context for the reader to understand Figure 2. I did not understand the implied meaning of [l_0, l_1]; I suspect that they are sizes, but I could not be sure.

~The results in Tables 1 and 2 show interesting improvements over the baselines, but there is an apparent mismatch between the performance of SAC reported in the paper (https://arxiv.org/pdf/1801.01290.pdf). The average performances of TD3 and SAC are lower in Tables 1 and 2 of this submission than in Figure 1 of the SAC paper. This mismatch is statistically possible, but it would be good for the authors to confirm this.~ I actually went back and compared Figure 9 in the submission's appendix with Figures 1 and 4 in the SAC paper, and the curves look similar.

The paper has sample efficiency in the title, but results and discussion in the main text talk about final performance only. The results in the appendix (Figure 9) are actually quite interesting, and we see some significant improvements with SAC in Ant. I suggest referring to these results in more detail in the main text (maybe mention in words what the results show and point to the appendix).

There are technical issues with notation that should be fixed. For example, some terms are not clearly defined:
* Blackboard Q. Sometimes it is a function mapping states and actions to real numbers, other times it is the space of these functions.
* \chi, \psi, \psi^-. These are very important for conveying the paper's main idea, but I had to fill in the details with guesses as I read the paper. I assumed that f_{\psi^-} would play the role of \chi_{\pi_i}, but I was not sure this was precisely right.

I could somewhat follow section 3.1., but I did not understand its purpose and I could not understand the purpose of the main result. My phrasing for Theorem 3 is "the representation at iteration t+1 reduces the policy evaluation error relative to the representation in iteration t+1" but how can we guarantee that the representation is indeed getting better? Ideally the representation would help in such a way that, as you iterate, the error of the best representation at that iteration is smaller than the error of the previous best with respect to the policy in the previous iteration.

However, I think section 5.3 makes up with empirical data for some of the issues with section 3.1. Overall I find interesting the discussion on the relationship between performance and the quality of the representation for fitting value functions, but I think this kind of discussion would benefit from a more directed study that fleshes it out.

For section 3.2, I would not see GOPE as a new method. The idea is natural and has been used in the past: Keep a replay buffer around, and keep adding data from past policies to it. The real contribution of this paper is the method to also encode the policy and make it part of the training data, and then use the encoded policy information as part of the representation provided to the policy.

Maybe the main point to make about section 3.2 is that we can take an off-policy learning algorithm and try to learn on all of the past data generated in the training process, and off-policy methods are designed to accommodate for this use-case, but that maybe we can do even better if we incorporate policy information into the data and use that in the off-policy method to improve performance.

I also suggest rewording "manner" as "method".

There is a claim that "policy representations optimized based on TD learning contain the most relevant features with value approximation". I find this claim very hard to back up depending on what you mean by relevant. If you have references to support this claim, please add them in too.

I found the claim in section 4.1 hard to follow as well. Please phrase the hypothesis more clearly, for example, "Our policy representations can ignore parameters associated with zero-output activations, without affecting the agent's performance." I also don't see how the average amount of parameter change allows us to make statements about the relevance of inactive neurons to the policy representations being proposed.

Is the content of Figure 3 essentially captured by Tables 1 and 2, modulo normalization and presentation?

Please fix the definition of Q^* in section 2.1.

There is an "it's" in paragraph 3 of section 3.2.

RanP in Tables 1 and 2 is referred to as RA in the main text. In my opinion it would be ok to remove the performance of a random agent from the comparisons.

---

### Official Review · Reviewer_4bv2 · 2023-10-30

**Soundness:** 2 fair
**Presentation:** 2 fair
**Contribution:** 2 fair
**Rating:** 5
**Confidence:** 4

**Summary:**

The paper proposes to learn value functions using low-dimensional policy representations. This can be understood as augmenting the value function space with policy representations, so that the value function can better generalize to unseen policies. The paper derives theoretical analysis for the approach and presents some empirical results which improve over baselines.

**Strengths:**

The paper presents some interesting analysis and some empirical results which show improvements over the baseline.

**Weaknesses:**

The idea of policy augmented value function is not new and as the authors have noted in multiple papers in previous literature. The theoretical analysis in the paper is still a bit weak and does not appear to capture the intuition behind the benefits of policy augmented value function. This puts into question the value of the analysis itself. The empirical results are interesting and show some promise, but would be in similar vein as the results displayed in prior work.

**Questions:**

=== **Them 3.3** ===

Thm 3.3 starts with a fairly strong assumption about the value learning error $f_t and $f_{t+1}$, this kind of puts into question the value of the result derived from the theorem. Imagine the situation where the difference between $Q^{\pi_t}$ and $Q^{\pi_{t+1}$ shrinks over time (e.g., when $\pi_t$ is converging to a fixed policy), then the assumption dictates that the value learning process becomes increasingly accurate. Note that since the learning target changes over time, this is not a supervised learning problem and hence the assumption on the error imposes non-trivial requirements on the learning system.

The $\leq$ shown in the error bound might also be not strong enough -- is there a chance that the error stays constant over time?

As a side note, directly assuming $|Q - Q^\pi|$ is not practical from a theoretical nor empirical perspective, because this error is not something that one can directly estimate from data. The convenient quantity that one can estimate would be Bellman errors.

=== **Theoretical intuition** ===

I feel the analysis in the paper does not capture the theoretical intuition behind the policy augmented value function, in that we should expect such an algorithm to work because of generalization. The analysis does not showcase how the value function generalizes to unseen policies. Thm 3.3 shows something in this flavor but with an assumption on how close $Q_{t+1}$ is to $Q^{\pi_{t+1}}$, which is too strong.

=== **Empirical result** ===

The empirical result appears promising but note that in much of the entries in Table 1-2, the standard deviations across runs are quite high and make it difficult to assess the actual statistical significance of the improvements. For example in SAC vs. SAC-GOPE on Hopper, both algorithms are similar in light of the big std coming from both sides of the algorithm, and one should not highlight SAC-GOPE as being better than SAC in a statistically significant sense.

The learning curves in Fig 8-9 look a bit suspicious in a few places: Why there is occassional dive in the learning curves of TD3 in Fig 8? why sometimes the TD3-GOPE and SAC-GOPE curves start in a slightly higher location than the baselines?

**Details Of Ethics Concerns:**

No concerns.

---

### Official Review · Reviewer_PEwK · 2023-10-31

**Soundness:** 3 good
**Presentation:** 3 good
**Contribution:** 2 fair
**Rating:** 3
**Confidence:** 4

**Summary:**

The authors suggest an efficient off-policy Reinforcement Learning approah through encoding the policy parameters. First, a new bellman operator, denoted as $\mathbb{T}$, is introduced to directly incorporate the policy representation into the value function. Subsequently, the authors introduce a Generalized Off-Policy Evaluation method (GOPE) that utilizes the proposed bellman operator and a simple policy representation learning method named Layer-wise Permutation-invariant Encoder with Dynamic Masking (LPE-DM).  The paper ultimately demonstrates that the proposed algorithm empirically outperforms prior off-policy methods and baseline algorithms across various tasks.

**Strengths:**

The paper proposes an interesting approach to augmenting value function inputs with low-dimensional policy representations and shows empirical improvements on various tasks. Additionally, the idea of dynamic masking in policy encoding is persuasive in policy representation, and the evidence supporting this argument is effectively presented.

**Weaknesses:**

The paper doesn't present a method that is fundamentally different from the approach in prior work[1], which concerns augementing value function inputs with policy representations. The proposed bellman operator seems rather trivial, as it appears to be a straightforward application of the policy-extended valu function $\mathbb{V}$ described in the paper[1]. The method of policy representation also closely mirrors the encoding of network parameters for OPR[1], with the exception of dynamic masking. While the paper clearly describes the concept and requirements of dynamic masking, the proposed method for it lacks intuitiveness. The Dynamic Masking (DM) technique masks the nodes of the policy network based on the node activity, which is determined by the number of parameter changes. However, the importance of parameters might not be soley defined by the frequency of their changes. Moreover, the paper does not address concerns related to complexity and memory. It should save the entire set of parameters from historical policy networks, deriving policy representation from these stored parameters. As a result, the proposed "GOPE" method is likely to require substantial memory and computational resources.

Several typographical errors were also identified in the paper:
   - In Section 2.1, the equation should be corrected from $Q^\ast (s,a) = \max_a Q^\pi (s,a)$ to $Q^\ast (s,a) = \max_\pi Q^\pi(s,a)$
   - In Equation (7), the notation should be revised from $i=N-1$ to $N-1$
   - In the Appendix, references to 'Table B' and 'Table C' should be revised to 'Table 3' and 'Table 4', respectively.

[1] Tang et al., What about Imputting Policy in Value Function: Policy Representation and Policy-Extended Value Function Approximator, AAAI22

**Questions:**

1. Figure 5 indicates that there is minimal performance difference between LPE and LPE-DM. Conducting additional ablation studies would be beneficial to underscore the necessity of Dynamic Masking (DM) in policy representation. An example could be an ablation study on the masking ratio $\eta$.

2. As previously discussed, the proposed DM, which relies on the number of parameter changes, lacks intuitiveness. Therefore, exploring alternative masking methods seems warranted. One approach to determine the importance of weights in a neural network is by employing Bayesian Neural Networks. Paper[1] utilizes a Bayesian Neural Network to identify the significance of weights. Could you consider experimenting with the Bayesian Neural Network for masking purposes?

3. Table 4 reveals that omitting the historical policy representation leads to enhanced performance in max-evaluation. This implies that the GOPE of the proposed method may not offer superior value approximation and generalization when finding the optimal policy. This observation is not addressed in the paper. Could you explain the underlying reasons behind this outcome?


[1] Kirkpatrick, James, et al. "Overcoming catastrophic forgetting in neural networks." Proceedings of the national academy of sciences 114.13 (2017): 3521-3526.

---

### Meta-Review · Area_Chair_hZXp · 2023-12-08

**Metareview:**

This paper proposes a method for off-policy RL via function approximation, where the Q function is conditioned on a representation of the learned policy. The proposed method crucially relies on a new Bellman operator, which is conditioned on the policy's representation. The paper uses the new Bellman operator in _Generalized Off-Policy Evaluation_ (GOPE), which works with any policy representation. The paper then proposes a particular policy representation learner it calls _Layer-wise Permutation-invariant Encoder with Dynamic Masking_ (LPE-DM). The paper concludes with an empirical evaluation of the proposed approach, comparing to existing off-policy learning algorithms.

The reviewers found the idea interesting, and the empirical results compelling. However, two reviewers questioned the novelty of the proposed method, given prior work (see reviews `PEwK ` and `4bv2` for details). Additionally, several reviewers found the presentation confusing, the reasoning difficult to follow, and lacking sufficient intuition for _why_ the proposed method should work.

The authors did not engage in the discussion period, so the reviewers' concerns went unresolved.

**Justification For Why Not Higher Score:**

There doesn't appear to be any disagreement amongst reviewers here; none of them thought this paper should be accepted.

**Justification For Why Not Lower Score:**

N/A

---

### Decision · Program_Chairs · 2024-01-16

Reject